# CausalEvolve: Towards Open-Ended Discovery with Causal Scratchpad

**Yongqiang Chen**[*1,2] **Chenxi Liu**[*3] **Zhenhao Chen**[1] **Tongliang Liu**[4,1] **Bo Han**[3] **Kun Zhang**[1,2]
[1]MBZUAI [2]Carnegie Mellon University [3]TMLR Group, Hong Kong Baptist University
[4]SAIC Centre, The University of Sydney
yqchen24@gmail.com   cscxliu@comp.hkbu.edu.hk

## Abstract

Evolve-based agent such as AlphaEvolve is one of the notable successes in using Large Language Models (LLMs) to build AI Scientists. These agents tackle open-ended scientific problems by iteratively improving and evolving programs, leveraging the prior knowledge and reasoning capabilities of LLMs. Despite the success, existing evolve-based agents lack targeted guidance for evolution and effective mechanisms for organizing and utilizing knowledge acquired from past evolutionary experience. Consequently, they suffer from decreasing evolution efficiency and exhibit oscillatory behavior when approaching known performance boundaries. To mitigate the gap, we develop `CausalEvolve`, equipped with a causal scratchpad that leverages LLMs to identify and reason about guiding factors for evolution. At the beginning, `CausalEvolve` first identifies outcome-level factors that offers complementary inspirations in improving the target objective. During the evolution, `CausalEvolve` also inspects surprise patterns during the evolution and abductive reasoning to hypothesize new factors, which in turn offer novel directions. Through comprehensive experiments, we show that `CausalEvolve` effectively improve the evolutionary efficiency and discovers better solutions in $4$ challenging open-ended scientific tasks.

## 1 Introduction

As large language model (LLMs) demonstrate increasing capabilities in complex and challenging reasoning tasks (Guo et al., 2025; Li et al., 2025e), the community seeks to build LLM-based agents to facilitate a number of downstream applications (Plaat et al., 2025). One of the most notable and promising applications is the AI Scientist agents (ZHENG et al., 2025), where the LLM-based agent is expected to automate the scientific discovery process ranging from conducting literature surveys (Wan et al., 2026), hypothesis generation (Khemakhem et al., 2020), data-driven analysis (Chan et al., 2024) to experiment design (Li et al., 2025c), etc. In fact, when incorporated into the agentic framework, LLMs have demonstrated great promise. Lu et al. (2024); Gottweis et al. (2025); Mitchener et al. (2025) show that LLMs can come up with new research hypotheses and proposals based on the existing literature and automate the full scientific discovery pipeline (Yamada et al., 2025). Recent advances in using LLMs to assist with scientific discovery shows LLMs can accelerate the idea iteration and deep literature search (Bubeck et al., 2025; Woodruff et al., 2026).

One of the most representative AI Scientist agents is the evolutionary coding agent, like AlphaEvolve (Novikov et al., 2025; Lange et al., 2025b). In the iterative evolutionary framework, LLMs demonstrate great capabilities in proposing, evaluating, and refining iteratively better solutions to a number of scientific problems (Sharma, 2025; Georgiev et al., 2025; Cheng et al., 2025). Despite the success, the evolution process in the existing

---

[*]These authors contributed equally.

frameworks is mainly driven by the evolution algorithm or derived from correlational studies. In contrast, human scientists can design *purposeful experiments* and *summarize scientific insights* from observational data (Kuhn & Hawkins, 1963; Kaelbling et al., 1998; Glymour). The gap that emerges between the uncontrolled evolutionary process of evolve-based agents and the guided discovery process of humans raises a challenging research question:

> *How can we develop evolution-based agents to perform guided scientific discovery like humans?*

To tackle the question, we resort to *causality*, which summarizes the practice of scientific discovery of humans (Spirtes et al., 2000b; Pearl, 2009). Essentially, scientific discovery is about revealing the underlying causal mechanism of the interested problem (Wallace, 1981; Glymour). Hence, we can formulate the evolution-based scientific discovery process as a Partially Observable Markov Decision Process (POMDP) (Kaelbling et al., 1998), where the agent needs to uncover the underlying causal mechanism through purposeful actions and interventions (Sec. 3). With the POMDP formulation, we demonstrate that accumulating and guiding the evolution with *causal knowledge* is crucial to both the efficiency and effectiveness of the discovery process. Without the incorporation of causality, the evolution can easily oscillate or get stuck at local optimal solutions.

To this end, we develop a new evolutionary AI Scientist framework, termed `CausalEvolve`, where we introduce a causal scratchpad to the evolution-based agent. The guidance provided by `CausalEvolve` is built upon the interventional factors identified before and during the evolution process. As the evolution-based agent primarily focuses on optimizing a target objective, such as the objective value of a combinatorial optimization problem or the accuracy of a machine learning problem (Lange et al., 2025b), `CausalEvolve` first identifies a set of *outcome-level factors* to provide complementary views of the target objective. During the evolution, `CausalEvolve` leverages a multi-arm bandit (MAB) to adaptively determine the desired intervention with respect to a selected outcome-level factor.

In addition, `CausalEvolve` also identifies *procedure-level factors* from the accumulated trials with LLMs (Liu et al., 2024). Intuitively, the procedure-level factors are useful interventions to the solutions that explain the objective value changes. For example, the optimization technique used to solve a combinatorial optimization problem. Nevertheless, some combinations of apparently useful factors may lead to decreased scores, which we term as "surprise patterns". Understanding and explaining the "surprise patterns" is critical to reveal new scientific insights (Wallace, 1981). Hence, `CausalEvolve` also performs abductive reasoning to come up with new factors and hypothesis that will be suggested to evaluate in the future experiments to better explain all the observed patterns (Douven, 2025).

Empirically, we show that `CausalEvolve` significantly improves the evolution efficiency and achieves better results compared to the existing state-of-the-art `ShinkaEvolve` (Lange et al., 2025b) across 4 open-ended discovery problems. Our contributions can be summarized as follows:

- We propose a theoretical formulation of evolution-based open-ended discovery, and demonstrate the necessity of causality (Sec. 3);
- We propose a new framework `CausalEvolve` to realize the accumulation and guidance of causal knowledge by identifying outcome-based and procedure-based factors;
- `CausalEvolve` is shown to improve both the evolution efficiency and effectiveness across 4 open-ended discovery problems.

## 2 RELATED WORK

**AI Scientist Agents.** With the significant advancement in LLM capacity and the development of Agentic system, there is a rising number of works on developing agents for helping scientific discoveries (Lu et al., 2024; Yamada et al., 2025; Gottweis et al., 2025). One research line is to automating the pipelines in scientific

activities, including literature review (Huang et al., 2025b), hypothesis generation (Li et al., 2024a; Yang et al., 2024; Wang et al., 2024; Yang et al., 2025), hypothesis verification (Li et al., 2024b; Huang et al., 2025a), and assistance in scientific reports (Liang et al., 2024). Another research line is to integrating the knowledge and reasoning ability of LLMs to conduct computational intensive evolution or iteration on specific scientific problems (Shojaee et al., 2025; Romera-Paredes et al., 2024; Novikov et al., 2025; Sharma, 2025; Lange et al., 2025a). There are also works on automated tabular data analysis with machine learning workflows (Zha et al., 2023; Li et al., 2023; Zhang et al., 2023; Li et al., 2025b), or embodied agents that can conduct real-world experiments (Roch et al., 2020; Zhu et al., 2022; Tom et al., 2024; Mandal et al., 2025). The impact of these lines of work has been made on scientific fields includes chemistry (Yang et al., 2026; Boiko et al., 2023), earth science (Feng et al., 2025), and biology (Swanson et al., 2025; Truhn et al., 2026).

**Causality for Scientific Discovery.** There has been a long history for the discussions on how to understand world through observations (Greenland et al., 1999; Spirtes et al., 2000a; Pearl, 2009). One research line is causal discovery for structured data, where algorithms are designed to learn directed acyclic graphs among the random variables as causal structure, including constrained-based methods (Spirtes et al., 1995; 2000a), methods with constrained functional (Shimizu et al., 2006; Zhang & Hyvarinen, 2012; Hoyer et al., 2008), non-stationarity (Malinsky & Spirtes, 2019; Huang et al., 2019; 2020; Liu & Kuang, 2023), the incorporation with multiple domain data (Huang et al., 2020; Yang et al., 2018; Brouillard et al., 2020; Mooij et al., 2020; Perry et al., 2022), and handling latent variables with the pure children assumption (Li et al., 2025d; Li & Liu, 2025). Recently, there are works to integrating causality with large language models. One direction is to empower the causal methods with the knowledge of LLMs, which includes constructing priors based on variable descriptions (Long et al., 2023; Li et al., 2024c), adjusting the causal structure searching process (Ban et al., 2023; Vashishtha et al., 2023; Jiralerspong et al., 2024), constructing structured variables out of unstructured data (Liu et al., 2025; Li et al., 2025a), and finding valid adjustment sets for treatment effect estimation (Dhawan et al., 2024; Liu et al., 2025; Sheth et al.). Another direction is to empower LLM-based agent with causal tools for tabular data analysis (Abdulaal et al., 2023; Khatibi et al., 2024; Shen et al., 2024; Wang et al., 2025a; Verma et al., 2025), revealing insights from data in an autonomous pipeline.

## 3 SCIENTIFIC DISCOVERY VIA OBJECTIVE OPTIMIZATION

### 3.1 FORMULATION OF SCIENTIFIC DISCOVERY

Scientific discovery aims to uncover the underlying scientific knowledge or the causal mechanisms from interactions with the world (Kuhn & Hawkins, 1963), which can be formulated as a Partially Observed Markov Decision Process (POMDP) (Kaelbling et al., 1998).

**Scientific knowledge.** The primary objective of an AI Scientist is to uncover the *underlying scientific knowledge* about the task-world, represented by a latent variable $\Theta_{\text{sci}} \in \Theta$, where $\Theta$ may encode causal structure, mechanisms, inductive biases, constraints, etc. Specifically, $\Theta_{\text{sci}} = \theta_{\text{sci}}$ can be parameterized as a Structural Causal Model (SCM) $\theta_{\text{sci}} = (\mathcal{G}, \mathcal{F}, P_U)$ (Spirtes et al., 2000a), where $\mathcal{G} = (V, E)$ is a directed graph whose nodes $V$ represent variables of interest and whose edges $E$ encode direct causal dependencies; $\mathcal{F} = \{f_v\}_{v \in V}$ is a collection of structural equations $v = f_v(\text{Pa}(v), u_v)$, where $\text{Pa}(v)$ denotes the parents of $v$ in $\mathcal{G}$ and $u_v$ is an exogenous noise variable; $P_U$ is a distribution over the exogenous variables $U = \{u_v\}_{v \in V}$.

**POMDP process.** Given $\theta_{\text{sci}}$, as shown in Fig. 1, the AI Scientist agent, implemented via the evolutionary coding framework such as `AlphaEvolve` (Novikov et al., 2025), will interact with the environment by proposing candidate programs $p_t \in \mathcal{P}$ (at turn $t$) to gain observations, $y_t = F(p_t, \theta_{\text{sci}})$, where $F : \mathcal{P} \times \Theta \to \mathbb{R}$ is the objective that the agent aims to optimize. Then, the scientific discovery process can be formulated as a POMDP $\mathcal{M} = (S, A, \Omega, \mathcal{T}, O, R, \gamma)$ with a static hidden parameter as $\theta_{\text{sci}}$ of the underlying scientific

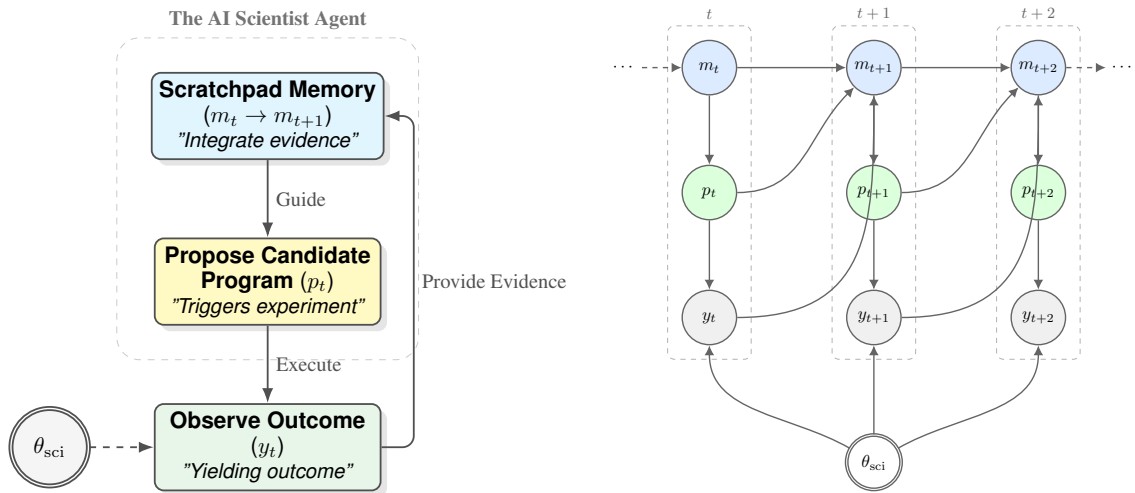

Figure 1: The iterative scientific discovery loop. **Left:** Conceptual flow of the agent. The agent maintains a scratchpad memory ($m$), proposes a program ($p$), and observes the outcome ($y$) which is constrained by the unknown world state ($\theta_{\mathrm{sci}}$). The outcome feeds back into the memory for the next step. **Right:** The diagram illustrates how the AI Scientist probes the unknown world state $\theta_{\mathrm{sci}}$. By proposing a candidate program $p_t$, the agent triggers an experiment yielding outcome $y_t$. This observation provides evidence about $\theta_{\mathrm{sci}}$, which is integrated into the agent's scratchpad memory $m_{t+1}$. Over time steps $t, t+1, \ldots$, this recurrent process allows the agent to navigate the performance landscape and converge towards optimal programs despite the static but unknown nature of $\theta_{\mathrm{sci}}$.

knowledge. The hidden state $s_t = \theta_{\mathrm{sci}}$ is the scientific knowledge $\theta_{\mathrm{sci}}$ that does not change over turns. The action is $a_t = p_t$ representing the choice of which program to evaluate. The observation $o_t = y_t$ is the evaluation outcome. The transition kernel $\mathcal{T}$ can be simply considered as identity, and the observation kernel is $O(o_t \mid s_t, a_t) = P(y_t \mid \theta_{\mathrm{sci}}, p_t)$. Given a finite experiment budget $T$, the agent chooses $p_0, \ldots, p_{T-1}$ and gain observations $y_0, \ldots, y_{T-1}$, so as to find $\hat{p} = \arg\max_p F(p, \theta_{\mathrm{sci}})$ and the scientific knowledge $\theta_{\mathrm{sci}}$.

**Evaluation as intervention on SCM.** Given the SCM parametrization of $\theta_{\mathrm{sci}}$, we can consider that a program $p \in \mathcal{P}$ is encoded as a particular configuration of design variables $X = x_p$. Then, $F$ can be implemented as

$$F(p; \theta_{\mathrm{sci}}) := \mathbb{E}\big[Y \,\big|\, \mathrm{do}(X = x_p), \theta_{\mathrm{sci}}\big], \tag{1}$$

i.e., the expected outcome under the intervention $\mathrm{do}(X = x_p)$ in the true causal model $\theta_{\mathrm{sci}}$. Typical implementations of $F$ can be the objective value of a combinatorial optimization problem, the efficiency of a kernel program, or the performance of a machine learning model (Novikov et al., 2025).

**Belief as a probability distribution over $\Theta$.** We define $b_t$ as the agent's Bayesian belief after history $h_t = \{(p_0, y_0), \ldots, (p_{t-1}, y_{t-1})\}$, i.e. a probability distribution on $\Theta$:

$$b_t(B) = \Pr(\Theta_{\mathrm{sci}} \in B \mid h_t, e), \qquad B \subseteq \Theta. \tag{2}$$

In the ideal Bayesian formalism, the belief $b_t(\theta)$ is a sufficient statistic for decision-making (Kaelbling et al., 1998). In practice, the AI Scientist maintains an internal belief, which is usually implemented as memory $m_t = \Phi(h_t)$ for some (possibly learnable) summarization function $\Phi$ (Lange et al., 2025a), to represent the *approximate representation* of its knowledge about $\theta_{\mathrm{sci}}$ and the landscape of $F(\cdot; \theta_{\mathrm{sci}})$. Each evaluation step $(p_t, y_t)$ thus updates $m_t$, which in turn updates the agent's effective belief about $\theta_{\mathrm{sci}}$. In this sense, each step *reveals part of the underlying scientific knowledge*, which in turn determines the next action $p_{t+1}$.

## 3.2 ESSENTIALITY OF CAUSAL KNOWLEDGE FOR AI SCIENTISTS

If the objective function $F$ is static universally, then with more experiment turns, the optimized solution $p_t$ and the agent's revealed scientific knowledge can also be applied universally. However, the observation from the evaluation is usually only given by a proxy knowledge $\theta_e$ about the scientific knowledge $\Theta_{\mathrm{sci}}$ at some specific environment $e \in \mathcal{E}$. For example, the performance of a machine learning model is usually assessed on finite samples from the test distribution, and there also exist distribution shifts from the test distribution when deploying the model in the real world (Quinonero-Candela et al., 2008). Different from $\Theta_{\mathrm{sci}}$ that characterizes the complete causal structure about the scientific problem, optimization under environment $\theta_e$ may introduce some spurious correlations that maximize the objective value $F_e$ (Chen et al., 2023). Therefore, without loss of generality, to retain the optimality of $\hat{p}$ beyond the source environment $e_{\mathrm{src}}$ to some target $e_{\mathrm{tgt}}$, it is essential to reveal the causal knowledge and answer causal questions for an AI Scientist.

**Definition 3.1** (Causal AI Scientist). *A Causal AI Scientist is an agent specified by: (i) a policy $\pi_t(\cdot \mid \theta_t, e_{\mathrm{src}})$ selecting $p_t$, (ii) a counterfactual / explanatory operator* CF, *that answer interventional queries $(e, p)$ via* $\mathrm{CF}(\theta_t; e, p)$ *as an "explanation" of predicted performance, where $\theta_t$ is the knowledge revealed at turn $t$.*

Without the revealing of the causal knowledge, the discovery process suffers from significant inefficiency and suboptimality issues. We discuss the two issues more concretely below.

**Evolutionary efficiency of Causal AI Scientist.** We begin by considering a static environment and finite $\mathcal{P} = \{p_1, \ldots, p_K\}$. For $\theta_{\mathrm{sci}}$, we assume each program $p$ has a known feature vector $x_p \in \mathbb{R}^d$ with $\|x_p\|_2 \leq 1$, and the unknown scientific parameter is a weight vector $w^\star \in \mathbb{R}^d$ and $F(p) = \langle x_p, w^\star \rangle$. Each evaluation returns a noisy observation $y_t = F(p_t) + \varepsilon_t$ where $\varepsilon_t \sim \mathcal{N}(0, \sigma^2)$ i.i.d.. A Causal AI Scientist in this environment can be implemented via estimating the $w^\star$ and optimizing for $\hat{p}$ from the history.

In addition, we also consider a black-box baseline that does not consider the interactions between the historical observations. It can be characterized as the following $\theta_{\mathrm{bb}} := \left\{ \mu : \mathcal{P} \to \mathbb{R} \right\}$ where each program has an unrelated unknown mean $F(p; \mu) = \mu(p)$, and $y_t = \mu(p_t) + \varepsilon_t$, where $\varepsilon_t$ is the same Gaussian noise.

**Theorem 3.2** (Informal). *Under the given environment, there exists a policy $\pi_{\mathrm{causal}}$ such that with probability at least $1 - \delta$, $F(\hat{p}; \theta_{\mathrm{sci}})$ obtains less than $2\epsilon$ error than the optimal value, with $O(d \log(K))$ turns; In contrast, the black-box baseline needs $O(K)$.*

The formal description of the sample efficiency issue and the proof are given in Appendix B. Theorem 3.2 shows that, when $K \gg d$, which is usually the case as the space for all programs is significantly larger than the underlying SCM, encoding (correct) causal structure yields an exponential (or at least multiplicative) gain in sample efficiency under finite budgets.

**Generalizability of Causal AI Scientist.** To show the necessity of capturing $\theta_{\mathrm{sci}}$, we have the following:

**Theorem 3.3.** *Consider the $e_{\mathrm{src}}, e_{\mathrm{tgt}} \in \mathcal{E}$ and $\theta_0, \theta_1 \in \Theta$ such that $F_{e_{\mathrm{src}}}(\cdot \mid p, \theta_0) = F_{e_{\mathrm{src}}}(\cdot \mid p, \theta_1)$ $\forall p \in \mathcal{P}$, and $\exists p, p' \in \mathcal{P}$ s.t. $F_{e_{\mathrm{tgt}}}(p; \theta_0) - F_{e_{\mathrm{tgt}}}(p'; \theta_0) \geq \Delta$ and $F_{e_{\mathrm{tgt}}}(p'; \theta_1) - F_{e_{\mathrm{tgt}}}(p; \theta_1) \geq \Delta$, for some $\Delta > 0$, then for any policy $\pi$ that can interact only with $e_{\mathrm{src}}$, there exists $i \in \{0, 1\}$ such that for every budget $T$, $\max_{p \in \mathcal{P}} F_{e_{\mathrm{tgt}}}(p; \theta_i) - F_{e_{\mathrm{tgt}}}(\hat{p}; \theta_i) \geq \Delta/2$.*

The formal description of the generalizability issue and the proof are given in Appendix C. Intuitively, Theorem 3.3 imply that if the source environment does not distinguish the corresponding $\theta_{\mathrm{sci}}$ among $\{\theta_0, \theta_1\}$, then the solution $\hat{p}$ solved given source environment is always suboptimal. In the real world, it is usually the case that two machine learning models will have similar performances under the public test benchmarks, but exhibit significantly different behaviors when generalizing to distributions from other environments.

## 4    CAUSAL SCRATCHPAD FOR EVOLUTIONARY CODING AGENT

Given the limitations shown in Sec. 3.2, it is essential to explicitly incorporate the causal knowledge into the evolutionary process. Hence, we present `CausalEvolve`, which incorporates a causal scratchpad to identify critical factors and exploit their causal relations with the objective variables to guide the evolution process. Specifically, we consider incorporating the outcome-level factors and the procedure-level factors to tackle the efficiency and the suboptimality issues, respectively.

### 4.1    OUTCOME-LEVEL FACTOR

Essentially, the underlying configurations of the program can be reflected and recognized from task-dependent, real-valued descriptors extracted from the *observable outcomes* of program execution. As shown in Theorem 3.2, intervening on the underlying configuration variables provides significantly higher sample efficiency.

**Factor construction.**    For a given task, a set of outcome-based factors $\mathbf{m} := (m_1, m_2, \ldots, m_K)$ is specified by LLMs before the evolution. An LLM would be prompted with the basic task description, which is the same as the system prompt used in evolution, and the expected output of each program, e.g., a list of coordinates, or an $n \times n$ matrix. For each of the outcome-based factors, the LLM would define the factor name and also a excitable code that maps the program output to the factor value. We list the outcome-based factors used in our tasks in Appendix D.

**Causal Planner with outcome-level factors.**    With outcome-based factors $\mathbf{m}$, we develop `CausalPlanner`. Specifically, we define the action space $\mathbf{A} := \cup_{m \in \mathbf{m}}\big\{(m, +1), (m, -1)\big\}$. When applying an action $(m, d)$, the existing programs would be sorted in descending order according to $m \times d$, and then the inspiration programs would be selected from the top of them. In $t$-th generation, after generating each child program from its parent and the inspiration programs with action $a \in \mathbf{A}$, the reward $R_a$ could be calculated. Let the $y_c$ be the child's main target that is to be maximized, and $v_t$ be the best-so-far value of the main target. We define the reward as $R_a := (y_c - \tau \cdot v_t)_+$, where $\tau \in (0, 1)$. We introduce this discounter $\tau$ because improving the best-so-far result could be a rare event, and therefore cannot be fairly estimated by only a few iterations. In practice, we alternate between exploration and exploitation: random actions are taken for $K$ iterations, followed by choosing the currently best action for the next $K$ iterations.

### 4.2    PROCEDURE-LEVEL FACTORS

To better capture important designs of the programs and uncover their associated causal knowledge, we also introduce procedure-level factors identified from the programs.

**Factor construction.**    We construct the procedure-level factors based on the `COAT` framework (Liu et al., 2025) that leverages LLMs to identify useful procedure factors from unstructured data. As LLMs are considered incapable of understanding causality, Liu et al. (2025) constructs feedback to regularize the identified factors by LLMs. Similarly, we prompt LLMs to identify factors that explain the performance differences of the performances of different programs. Then, `CausalEvolve` estimates an approximated average treatment effect of different factors with respect to the target objective value to provide a holistic view of the usefulness of the identified procedure-level factors. Due to the limited sample size and the existence of hidden confounders, the estimated treatment effects may contain biases, while empirically, we do not need an accurate estimation, but order-preserved quantities to provide insights.

**Abductive reasoning.**    As mainly explaining the performance differences is insufficient for revealing all factors, we also incorporate a surprise detection module and leverage LLMs to perform abductive reasoning

on the potentially existing factors and hypotheses that explain the surprise patterns (Douven, 2025). The detection of surprise patterns relies on the estimated treatment effects. Since the estimation can contain biases, we focus on detecting significant shifts in the estimated effects, including the signal inverses, i.e., a positively correlated factor produces negative effects, and significant quantity shifts, i.e., a minor correlated factor produces negative effects. By explaining the surprise patterns, we are able to find the underlying confounder and better reveal the underlying $\theta_{\text{sci}}$.

## 5 EXPERIMENTS

### 5.1 EXPERIMENTAL SETTING

**Baselines.** We mainly compare `CausalEvolve` with the state-of-the-art evolve-based agent `ShinkaEvolve` (Lange et al., 2025a) that produces the best or competitive results as `AlphaEvolve` (Novikov et al., 2025) in an sample-efficient manner. As `ShinkaEvolve` also incorporates a memory module to summarize the insights from $h_t$, we also consider two additional variants, `CausalPlanner` with meta summary module from `ShinkaEvolve`, and `COAT`, to ablate the effects of two modules in `CausalEvolve`. For the LLMs, we fix to using `Grok-4.1-fast-reasoning` (xAI, 2025) for fair comparisons.

**Tasks.** We evaluate our framework on four scientific discovery tasks that require optimizing code for different objectives:

**Hadamard Matrix** ($n = 29$). The goal is to construct an $n \times n$ matrix $H$ with entries in $\{\pm 1\}$ that maximizes the absolute determinant $|\det(H)|$. For $n = 29$, the best-known solution achieves $|\det(H)| = 2^{28} \cdot 7^{12} \cdot 320$, which we use to normalize scores to $[0, 1]$ for comparability with prior work (Wang et al., 2025b). This discrete optimization problem requires balancing matrix properties including row orthogonality, element balance, and determinant magnitude.

**Second Autocorrelation Inequality.** We seek a step function $f : [-1, 1] \to \mathbb{R}_{\geq 0}$ (discretized into $n = 256$ steps) that minimizes the ratio
$$R(f) = \frac{\|f * f\|_2^2}{\|f * f\|_1 \|f * f\|_\infty},$$
where $f * f$ denotes linear autoconvolution. The optimal value $R(f) \geq 1.1547\ldots$ remains an open conjecture. This continuous optimization task requires carefully shaping the function's smoothness, concentration, and sparsity.

**Circle Packing** ($N = 26$). The objective is to place $N$ circles with radii $r_i$ and centers $C_i = (x_i, y_i)$ in a unit square $[0, 1]^2$ such that: (i) no circles overlap ($\|C_i - C_j\| \geq r_i + r_j$ for all $i \neq j$), (ii) all circles remain within the square ($r_i \leq C_i^x, C_i^y \leq 1 - r_i$), and (iii) the sum of radii $\sum_i r_i$ is maximized. This geometric optimization task requires spatial reasoning about density, distribution, and boundary constraints.

**AIME Mathematical Problem Solving.** We evaluate on the 2024 American Invitational Mathematics Examination (AIME), a challenging competition consisting of 15 problems requiring integer answers in $[000, 999]$. The task is to build an LLM-based agent that solves these problems efficiently. Performance is measured by accuracy, while auxiliary metrics track format compliance (e.g., \boxed{} format), cost efficiency, and stability across problems.

**Evaluation metrics.** We run every method using 3 random seeds $(1, 2, 3)$ to accommodate the randomness. To compare the efficiency and the optimality, we inspect the stepwise averaged results as well as the best result from the 3 runs, at 4 intermediate steps. Given the difficulty of different tasks, we inspect steps $50, 100, 150, 200$ for Second Autocorrelation Inequality and Circle Packing, steps $20, 40, 80, 100$ for Hadamard Matrix, and steps $20, 40, 60, 80$ for AIME agent.

Table 1: **Main results across four scientific discovery tasks.** Performance is reported at training steps 1 through 4. For each step, we report the mean performance (Mean) and the best-so-far value (Best). All tasks are maximization objectives.

| Task | Method | Grok-4.1-FR | | | | | | | |
|---|---|---|---|---|---|---|---|---|---|
| | | Step 1 | | Step 2 | | Step 3 | | Step 4 | |
| | | Mean | Best | Mean | Best | Mean | Best | Mean | Best |
| **Hadamard Matrix** (↑) | ShinkaEvolve | 0.495 | 0.533 | 0.521 | 0.540 | 0.521 | 0.540 | 0.521 | 0.540 |
| | CausalPlanner (Meta) | 0.556 | 0.573 | 0.567 | 0.573 | 0.567 | 0.573 | 0.567 | 0.573 |
| | COAT | 0.503 | 0.519 | 0.514 | 0.543 | 0.521 | 0.552 | 0.532 | 0.561 |
| | CausalEvolve | 0.542 | 0.574 | 0.550 | 0.574 | 0.563 | 0.576 | **0.568** | **0.576** |
| **Second Autocorr. Inequality** (↑) | ShinkaEvolve | 0.723 | 0.724 | 0.729 | 0.739 | 0.735 | 0.749 | 0.737 | 0.751 |
| | CausalPlanner (Meta) | 0.730 | 0.745 | 0.734 | 0.749 | 0.735 | 0.750 | 0.736 | 0.750 |
| | COAT | 0.753 | 0.770 | 0.771 | 0.783 | 0.773 | 0.783 | 0.783 | 0.786 |
| | CausalEvolve | 0.781 | 0.800 | 0.783 | 0.805 | 0.790 | 0.809 | **0.793** | **0.809** |
| **Circle Packing** (↑) | ShinkaEvolve | 2.342 | 2.431 | 2.342 | 2.431 | 2.400 | 2.435 | 2.479 | 2.500 |
| | CausalPlanner (Meta) | 2.348 | 2.541 | 2.358 | 2.541 | 2.456 | 2.541 | 2.456 | 2.541 |
| | COAT | 2.183 | 2.261 | 2.238 | 2.292 | 2.436 | 2.560 | 2.456 | **2.568** |
| | CausalEvolve | 2.106 | 2.295 | 2.216 | 2.370 | 2.385 | 2.516 | **2.476** | 2.564 |
| **AIME Agent** (↑) | ShinkaEvolve | 33.33 | 33.33 | 34.44 | 36.67 | 34.44 | 36.67 | 34.44 | 36.67 |
| | CausalPlanner (Meta) | 34.44 | 36.67 | 35.56 | 36.67 | 36.67 | 40.00 | 36.67 | 40.00 |
| | COAT | 37.78 | 43.33 | 37.78 | 43.33 | 37.78 | 43.33 | 38.89 | **43.33** |
| | CausalEvolve | 33.33 | 36.67 | 38.89 | 40.00 | 38.89 | 40.00 | 38.89 | 40.00 |

## 5.2 EXPERIMENTAL RESULTS

The results of the experiments are given in Table 1. From the results, we can find that across all tasks, CausalEvolve produce significantly better averaged results than ShinkaEvolve across different tasks and steps, demonstrating the effectiveness of CausalEvolve. Notably, in AIME, CausalEvolve achieves 38.89% results based on the same scaffolding agent as in ShinkaEvolve. While in the original paper of ShinkaEvolve, even with a more sophisticated ensemble of multiple frontier reasoning models, ShinkaEvolve can only achieve a performance of 34.4%, demonstrating the effectiveness of CausalEvolve in breaking the state-of-the-art results in the open-ended discovery.

When comparing different variants and CausalEvolve, we can find that, across 4 tasks, CausalEvolve maintain the overall best performances, verifying that each module is essential to the success of CausalEvolve. Interestingly, in the majority of tasks, COAT can already produce an impressive best result, demonstrating the effectiveness of procedure-level factors for optimality. When comparing results with and without CausalPlanner, we can also find that with CausalPlanner, we can achieve better results already at early steps, demonstrating the effectiveness of outcome-based factors in sample efficiency.

## 6 CONCLUSIONS

In this work, we studied the evolutionary coding agent for scientific discovery. With the POMDP formulation of the discovery process, we demonstrate the necessity of incorporating causal knowledge. Then, we propose CausalEvolve that uses a causal scratchpad to identify and exploit outcome-based and procedure-based factors and the associated causal knowledge to guide the evolution process. Empirical results with 4 discovery tasks verified the improved efficiency and optimality of CausalEvolve.

ACKNOWLEDGMENTS

We thank the reviewers for their constructive comments and suggestions.

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

## LLM USE STATEMENT

From the research side, this work studies the use of LLMs for automated scientific discovery. From the paper writing side, we use LLMs to assist with improving the writing of this work.

## ETHICS STATEMENT

We study using LLMs to automate scientific discovery that will benefit the whole humanity and society. This work does not involve human subjects or personally identifiable information beyond public benchmarks used under their licenses.

## A  ADDITIONAL TECHNICAL DETAILS

### A.1  NOTATION

Table 2: Notation used in the formulation and theorems.

| Symbol | Meaning |
|---|---|
| $\mathcal{P}$ | Program / pipeline / model space (candidate designs) |
| $K$ | Number of candidate programs, $K := |\mathcal{P}|$ (finite in Theorem 1) |
| $p \in \mathcal{P}$ | A program to evaluate (action) |
| $\mathcal{X}$ | Design-variable space (encoding of programs) |
| $x_p \in \mathcal{X}$ | Encoding of program $p$ (e.g., design variables $X = x_p$) |
| $\Theta$ | Hypothesis space of scientific knowledge (e.g., SCMs / mechanisms) |
| $\Theta_{\mathrm{sci}}$ | Latent RV taking values in $\Theta$ (Bayesian view) |
| $\theta^\star \in \Theta$ | True (fixed but unknown) scientific knowledge instance (realization) |
| $\mu_0$ | Prior over $\Theta$ (i.e., $\Theta_{\mathrm{sci}} \sim \mu_0$) |
| $\mathcal{E}$ | Environment / protocol index set (evaluation regimes, deployments) |
| $e \in \mathcal{E}$ | Environment index; $e_{\mathrm{src}}$ source, $e_{\mathrm{tgt}}$ target |
| $F_e(p; \theta)$ | True performance in env $e$ (scalar objective) |
| $P_e(\cdot \mid p, \theta)$ | Observation model (likelihood) for evaluator output in env $e$ |
| $y_t$ | Observed evaluator outcome at round $t$ |
| $h_t$ | History $\{(p_0, y_0), \ldots, (p_{t-1}, y_{t-1})\}$ |
| $b_t$ | Bayesian belief/posterior over $\theta$: $b_t(\cdot) = \Pr(\Theta_{\mathrm{sci}} \in \cdot \mid h_t, e_{\mathrm{src}})$ |
| $T$ | Evaluation budget / horizon (number of program evaluations) |

### A.2  RANDOM VARIABLE, SPACE, AND REALIZATION (TO AVOID NOTATION CONFUSION)

We use the following (standard) convention.

**(i) Hypothesis space.** $\Theta$ is a set that contains all candidate scientific-knowledge hypotheses.

**(ii) True but unknown instance.** The real world is governed by a fixed but unknown $\theta^\star \in \Theta$.

**(iii) Bayesian view (optional but convenient).** A Bayesian agent models uncertainty by treating $\theta^\star$ as a realization of a latent random variable $\Theta_{\mathrm{sci}}$ with prior $\mu_0$, i.e. $\Theta_{\mathrm{sci}} \sim \mu_0$ and $\theta^\star$ is one draw from it. The belief $b_t$ is simply the posterior distribution of $\Theta_{\mathrm{sci}}$ after seeing history $h_t$.

**(iv) Does scientific knowledge change across environments?** In our formulation, the *underlying* scientific knowledge $\theta^\star$ is static across rounds. Different environments $e \in \mathcal{E}$ represent different evaluation/deployment protocols (distribution shifts, constraint changes, measurement noise, private vs public tests, etc.). Formally,

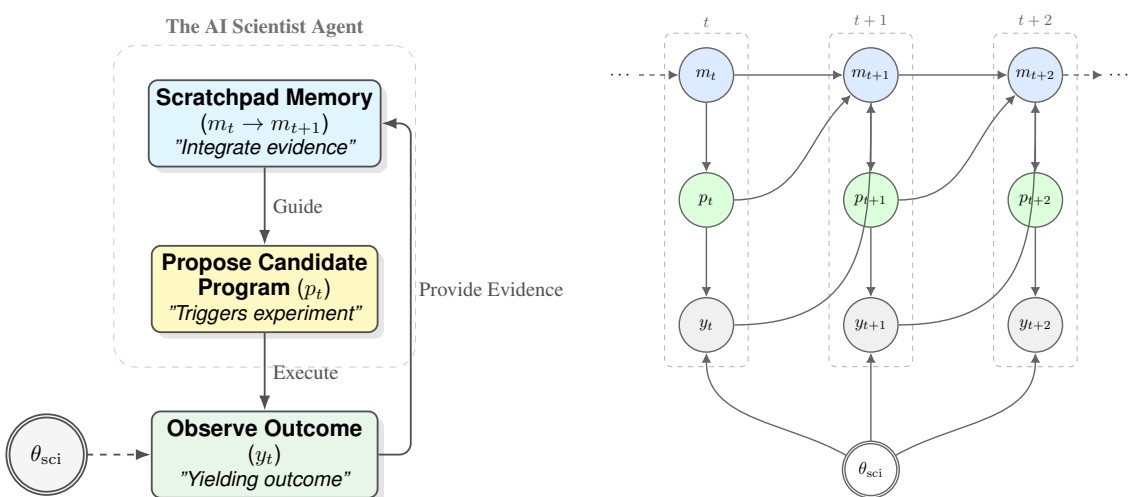

Figure 2: The iterative scientific discovery loop. **Left:** Conceptual flow of the agent. The agent maintains a scratchpad memory ($m$), proposes a program ($p$), and observes the outcome ($y$) which is constrained by the unknown world state ($\theta_{\mathrm{sci}}$). The outcome feeds back into the memory for the next step. **Right:** The diagram illustrates how the AI Scientist probes the unknown world state $\theta_{\mathrm{sci}}$. By proposing a candidate program $p_t$, the agent triggers an experiment yielding outcome $y_t$. This observation provides evidence about $\theta_{\mathrm{sci}}$, which is integrated into the agent's scratchpad memory $m_{t+1}$. Over time steps $t, t+1, \ldots$, this recurrent process allows the agent to navigate the performance landscape and converge towards optimal programs despite the static but unknown nature of $\theta_{\mathrm{sci}}$.

environments affect either the true performance map $F_e(\cdot; \theta)$ and/or the observation kernel $P_e(\cdot \mid p, \theta)$, while $\theta^\star$ itself remains the same hidden instance.

### A.3  EVALUATOR AS AN OBSERVATION MODEL (COVERS DETERMINISTIC AND STOCHASTIC EVALUATORS)

Fix an environment $e \in \mathcal{E}$. When the agent evaluates program $p$, it receives an observation $y \in \mathcal{Y}$ drawn from

$$y \sim P_e(\cdot \mid p, \theta^\star),$$

where $P_e(\cdot \mid p, \theta)$ is a conditional distribution on $\mathcal{Y}$.

**Deterministic evaluator.** A deterministic evaluator is the special case where there exists a function $g_e$ such that

$$P_e(\cdot \mid p, \theta) = \delta_{g_e(p;\theta)}(\cdot), \quad \text{i.e.,} \quad y = g_e(p; \theta^\star) \text{ a.s.}$$

In many program-evolution settings, the evaluator is designed to deterministically check validity and compute an objective score (e.g., via a verifier and a scoring routine).

**Stochastic/noisy evaluator.** A common instantiation is additive noise:

$$y = F_e(p; \theta^\star) + \varepsilon, \qquad \varepsilon \sim \mathcal{N}(0, \sigma^2),$$

but our proofs only rely on the specific Gaussian form in Theorem 1.

### A.4 BELIEF AND BAYES UPDATE: KERNEL FORM AND UNDERGRADUATE-FRIENDLY SPECIAL CASES

Let $h_t = \{(p_0, y_0), \ldots, (p_{t-1}, y_{t-1})\}$ be the history. The Bayesian belief (posterior) is

$$b_t(B) = \Pr(\Theta_{\mathrm{sci}} \in B \mid h_t, e_{\mathrm{src}}), \qquad B \subseteq \Theta.$$

**General Bayes update (kernel form).** After choosing $p_t$ and observing $y_t$ in $e_{\mathrm{src}}$, the posterior is

$$b_{t+1}(B) = \frac{\int_B P_{e_{\mathrm{src}}}(dy_t \mid p_t, \theta)\, b_t(d\theta)}{\int_\Theta P_{e_{\mathrm{src}}}(dy_t \mid p_t, \theta)\, b_t(d\theta)}. \tag{3}$$

**Finite hypothesis space (sum form).** If $\Theta = \{\theta_1, \ldots, \theta_N\}$ is finite and the likelihood has a pmf $P_{e_{\mathrm{src}}}(y_t \mid p_t, \theta_i)$, then

$$b_{t+1}(\theta_i) = \frac{b_t(\theta_i)\, P_{e_{\mathrm{src}}}(y_t \mid p_t, \theta_i)}{\sum_{j=1}^N b_t(\theta_j)\, P_{e_{\mathrm{src}}}(y_t \mid p_t, \theta_j)}.$$

**Continuous hypothesis space (density form).** If $P_{e_{\mathrm{src}}}(dy \mid p, \theta)$ has a density $p_{e_{\mathrm{src}}}(y \mid p, \theta)$, then

$$b_{t+1}(\theta) = \frac{b_t(\theta)\, p_{e_{\mathrm{src}}}(y_t \mid p_t, \theta)}{\int_\Theta b_t(\theta')\, p_{e_{\mathrm{src}}}(y_t \mid p_t, \theta')\, d\theta'}.$$

**Deterministic evaluator (indicator/filter form).** If $y = g_{e_{\mathrm{src}}}(p; \theta)$ deterministically, then the update becomes

$$b_{t+1}(d\theta) \ \propto \ \mathbf{1}\{g_{e_{\mathrm{src}}}(p_t; \theta) = y_t\}\, b_t(d\theta),$$

i.e. the posterior is the prior restricted to hypotheses consistent with the observed outcome.

## B PROOF OF THEOREM 3.2 (STATIC SAMPLE-EFFICIENCY GAP)

Throughout this section we fix a *single* static environment (drop $e$ from notation), and assume $\mathcal{P} = \{p_1, \ldots, p_K\}$ is finite.

### B.1 PROTOCOL AND PERFORMANCE CRITERION

**Experiment–then–commit protocol.** A policy $\pi$ interacts for $T$ rounds. At each round $t = 0, \ldots, T-1$ it selects a program $p_t \in \mathcal{P}$ (possibly randomized) based on the past history $h_t$, then observes $y_t \in \mathbb{R}$. After $T$ evaluations it outputs a final recommendation $\hat{p} \in \mathcal{P}$.

**Simple regret.** Let $f(p)$ denote the true mean performance of program $p$ in this environment. Define the (random) simple regret

$$\mathrm{SR}_T := \max_{p \in \mathcal{P}} f(p) \ - \ f(\hat{p}). \tag{4}$$

$(\epsilon, \delta)$**-correctness (uniform).** Fix $\epsilon > 0$ and $\delta \in (0, 1)$. We say a policy $\pi$ is $(\epsilon, \delta)$-*correct uniformly on a hypothesis class* $\mathcal{H}$ if for every instance in $\mathcal{H}$,

$$\Pr\left(\mathrm{SR}_T \leq \epsilon\right) \geq 1 - \delta.$$

"Uniformly" means the guarantee must hold for *all* instances in the class, not only on average.

## B.2 TWO HYPOTHESIS CLASSES

**(1) Structured (causal/scientific) linear class.** Each program $p$ has a known feature vector $x_p \in \mathbb{R}^d$ with $\|x_p\|_2 \le 1$. The unknown instance is a weight vector $w^\star \in \mathbb{R}^d$ and

$$f(p) = \langle x_p, w^\star \rangle. \tag{5}$$

Observations follow a Gaussian noise model

$$y_t = f(p_t) + \varepsilon_t, \qquad \varepsilon_t \sim \mathcal{N}(0, \sigma^2) \text{ i.i.d.} \tag{6}$$

Assume there exist $d$ basis programs $p^{(1)}, \ldots, p^{(d)}$ whose feature vectors are the standard basis:

$$x_{p^{(i)}} = e_i, \qquad i = 1, \ldots, d. \tag{7}$$

**(2) Unstructured black-box class (baseline).** The unknown instance is an arbitrary vector of means

$$\mu = (\mu_1, \ldots, \mu_K) \in \mathbb{R}^K, \qquad f(p_i) = \mu_i,$$

and observations are

$$y_t = \mu_{I_t} + \varepsilon_t, \qquad \varepsilon_t \sim \mathcal{N}(0, \sigma^2) \text{ i.i.d.}, \tag{8}$$

where $I_t \in \{1, \ldots, K\}$ is the index of the chosen program $p_t = p_{I_t}$. Crucially, there is *no assumed relation* between $\mu_i$ and $\mu_j$ for $i \ne j$.

## B.3 FORMAL STATEMENT AND PROOF

**Theorem B.1** (Formal version of Theorem 3.2)**.** *Fix $\epsilon > 0$ and $\delta \in (0, 1/4)$.*

1. *(**Upper bound under the structured linear class**). Under equation 5–equation 7 and equation 6, there exists a policy $\pi_{\text{lin}}$ such that*

$$\Pr\left(\text{SR}_T \le 2\epsilon\right) \ge 1 - \delta \quad \text{whenever} \quad T \ge 2d \frac{\sigma^2}{\epsilon^2} \log\left(\frac{2K}{\delta}\right).$$

2. *(**Lower bound for the unstructured black-box class**). For the black-box class equation 8, any policy that is $(\epsilon, \delta)$-correct uniformly for all $\mu \in \mathbb{R}^K$ must satisfy*

$$T \ge (K - 1) \frac{\sigma^2}{8\epsilon^2} \log\left(\frac{1}{2\delta}\right).$$

*Proof.* We prove the two parts separately.

**Part (1): constructive upper bound (estimate $w^\star$ then commit).** Evaluate each basis program $p^{(i)}$ exactly $n$ times (total $T = nd$). Let $y_1^{(i)}, \ldots, y_n^{(i)}$ be the observations for basis $i$, and define

$$\hat{w}_i := \frac{1}{n} \sum_{j=1}^{n} y_j^{(i)}.$$

By equation 5–equation 7, $f(p^{(i)}) = w_i^\star$. By equation 6, $\hat{w}_i \sim \mathcal{N}(w_i^\star, \sigma^2/n)$ and these coordinates are independent.

Define for any program $p$:

$$\hat{f}(p) := \langle x_p, \hat{w} \rangle, \qquad \hat{w} = (\hat{w}_1, \ldots, \hat{w}_d).$$

Then

$$\widehat{f}(p) - f(p) = \langle x_p, \hat{w} - w^\star \rangle \sim \mathcal{N}\Big(0, \ \frac{\sigma^2}{n}\|x_p\|_2^2\Big),$$

so since $\|x_p\|_2 \leq 1$,

$$\Pr\big(|\widehat{f}(p) - f(p)| \geq \epsilon\big) \leq 2\exp\Big(-\frac{n\epsilon^2}{2\sigma^2}\Big).$$

Union bound over $K$ programs gives

$$\Pr\Big(\max_{p \in \mathcal{P}}|\widehat{f}(p) - f(p)| \geq \epsilon\Big) \leq 2K\exp\Big(-\frac{n\epsilon^2}{2\sigma^2}\Big).$$

Choose

$$n \ \geq \ 2\frac{\sigma^2}{\epsilon^2}\log\Big(\frac{2K}{\delta}\Big),$$

so that with probability at least $1 - \delta$ we have $\max_p |\widehat{f}(p) - f(p)| \leq \epsilon$.

Now output $\hat{p} := \arg\max_{p \in \mathcal{P}} \widehat{f}(p)$. Let $p^\star := \arg\max_p f(p)$. On the above high-probability event,

$$f(p^\star) - f(\hat{p}) \leq \big(f(p^\star) - \widehat{f}(p^\star)\big) + \big(\widehat{f}(\hat{p}) - f(\hat{p})\big) \leq \epsilon + \epsilon = 2\epsilon.$$

Thus $\Pr(\mathrm{SR}_T \leq 2\epsilon) \geq 1 - \delta$ for $T = nd$ as stated.

**Part (2): lower bound for the black-box class.** We construct $K$ hard instances and lower bound any uniformly $(\epsilon, \delta)$-correct policy.

Let the programs be $p_1, \ldots, p_K$. Define a base instance $\mu^{(0)} \in \mathbb{R}^K$:

$$\mu_1^{(0)} = 0, \qquad \mu_i^{(0)} = -2\epsilon \ (i = 2, \ldots, K).$$

For each $i \in \{2, \ldots, K\}$, define an alternative instance $\mu^{(i)}$:

$$\mu_1^{(i)} = 0, \qquad \mu_i^{(i)} = +2\epsilon, \qquad \mu_j^{(i)} = -2\epsilon \ (j \notin \{1, i\}).$$

Under $\mu^{(0)}$, the unique best program is $p_1$, and choosing any $p_i$ with $i \geq 2$ incurs regret $2\epsilon > \epsilon$. Under $\mu^{(i)}$, the unique best program is $p_i$, and choosing $p_1$ incurs regret $2\epsilon > \epsilon$.

Let $P_0$ be the distribution of the full transcript $\mathcal{T} := (p_{0:T-1}, y_{0:T-1}, \hat{p})$ under $\mu^{(0)}$, and $P_i$ the analogous distribution under $\mu^{(i)}$. Uniform $(\epsilon, \delta)$-correctness implies

$$P_0(\hat{p} = p_1) \geq 1 - \delta, \qquad P_i(\hat{p} = p_1) \leq \delta \quad (i = 2, \ldots, K).$$

**Step 1: a KL lower bound from an event.** For any event $A$ and distributions $P, Q$, one has

$$\mathrm{KL}(P\|Q) \geq P(A)\log\frac{P(A)}{Q(A)} + (1 - P(A))\log\frac{1 - P(A)}{1 - Q(A)}.$$

Apply it with $A = \{\hat{p} = p_1\}$, $P = P_0$, $Q = P_i$. Let $p := P_0(A) \geq 1 - \delta$ and $q := P_i(A) \leq \delta$. For $\delta \in (0, 1/4)$ this yields

$$\mathrm{KL}(P_0\|P_i) \ \geq \ \log\Big(\frac{1}{2\delta}\Big). \tag{9}$$

**Step 2: compute** $\mathrm{KL}(P_0\|P_i)$ **via number of pulls of arm** $i$. Under $\mu^{(0)}$ and $\mu^{(i)}$, the policy is identical; only observations when playing $p_i$ differ:

$$y \sim \mathcal{N}(-2\epsilon, \sigma^2) \text{ under } \mu^{(0)}, \qquad y \sim \mathcal{N}(+2\epsilon, \sigma^2) \text{ under } \mu^{(i)}.$$

For Gaussians with equal variance, $\mathrm{KL}(\mathcal{N}(m_0, \sigma^2) \| \mathcal{N}(m_1, \sigma^2)) = \frac{(m_0 - m_1)^2}{2\sigma^2}$, so each pull of $p_i$ contributes $\mathrm{KL}\ \frac{(4\epsilon)^2}{2\sigma^2} = \frac{8\epsilon^2}{\sigma^2}$.

Let $N_i$ be the (random) number of times $p_i$ is evaluated in $T$ rounds. Additivity of log-likelihood ratios over independent Gaussian samples yields

$$\mathrm{KL}(P_0 \| P_i) = \frac{8\epsilon^2}{\sigma^2}\, \mathbb{E}_{P_0}[N_i]. \tag{10}$$

**Step 3: conclude the lower bound on $T$.** Combine equation 9 and equation 10:

$$\mathbb{E}_{P_0}[N_i] \ \geq\ \frac{\sigma^2}{8\epsilon^2} \log\!\Big(\frac{1}{2\delta}\Big), \qquad i = 2, \ldots, K.$$

Summing over $i = 2, \ldots, K$ gives

$$T = \sum_{i=1}^{K} N_i \ \geq\ \sum_{i=2}^{K} \mathbb{E}_{P_0}[N_i] \ \geq\ (K-1)\frac{\sigma^2}{8\epsilon^2} \log\!\Big(\frac{1}{2\delta}\Big).$$

This completes the proof. $\qquad\qquad\qquad\qquad\qquad\qquad\qquad\qquad\qquad\qquad\qquad\qquad\qquad$ $\square$

**Remark (deterministic evaluator).** If $\sigma = 0$, the structured linear class can recover $w^\star$ exactly from $d$ basis evaluations and achieve $\mathrm{SR}_T = 0$, while in the unstructured black-box class a uniform worst-case guarantee requires evaluating all $K$ programs at least once.

**Reference for the black-box lower bound.** The above is a standard change-of-measure/KL argument for best-arm identification in $K$-armed Gaussian bandits (e.g., see classical treatments of best-arm identification lower bounds).

## C    PROOF OF THEOREM 3.3 (NON-IDENTIFIABILITY UNDER ENVIRONMENT SHIFTS)

### C.1    SETUP: SOURCE INTERACTION, TARGET EVALUATION, AND TARGET REGRET

The agent can only interact with the *source* environment $e_{\mathrm{src}}$:

$$y_t \sim P_{e_{\mathrm{src}}}(\cdot \mid p_t, \theta^\star).$$

After $T$ rounds it outputs a final program $\hat{p}$. Performance is judged in a *target* environment $e_{\mathrm{tgt}}$ via $F_{e_{\mathrm{tgt}}}(p; \theta^\star)$. Define the target (simple) regret:

$$\mathrm{GR}_T(\theta^\star) := \max_{p \in \mathcal{P}} F_{e_{\mathrm{tgt}}}(p; \theta^\star) - F_{e_{\mathrm{tgt}}}(\hat{p}; \theta^\star).$$

### C.2    FORMAL STATEMENT AND PROOF

**Theorem C.1** (Non-identifiability barrier under shifts). *Fix $e_{\mathrm{src}}, e_{\mathrm{tgt}} \in \mathcal{E}$. Assume there exist two hypotheses $\theta_0, \theta_1 \in \Theta$ such that:*

*(Source indistinguishability)* $\quad P_{e_{\mathrm{src}}}(\cdot \mid p, \theta_0) = P_{e_{\mathrm{src}}}(\cdot \mid p, \theta_1), \quad \forall p \in \mathcal{P}.$ $\qquad\qquad$ (11)

*(Target optimal action flips with margin $\Delta$)* $\quad \exists\, p_0, p_1 \in \mathcal{P}$ *and* $\Delta > 0$ *s.t.* $\qquad\qquad$ (12)

$$p_0 \in \arg\max_{p \in \mathcal{P}} F_{e_{\mathrm{tgt}}}(p; \theta_0), \quad p_1 \in \arg\max_{p \in \mathcal{P}} F_{e_{\mathrm{tgt}}}(p; \theta_1),$$

$$F_{e_{\mathrm{tgt}}}(p_0; \theta_0) - F_{e_{\mathrm{tgt}}}(p; \theta_0) \geq \Delta,\ \forall p \neq p_0, \quad F_{e_{\mathrm{tgt}}}(p_1; \theta_1) - F_{e_{\mathrm{tgt}}}(p; \theta_1) \geq \Delta,\ \forall p \neq p_1. \tag{13}$$

*Then for any policy $\pi$ that can interact only with $e_{\mathrm{src}}$, there exists $i \in \{0,1\}$ such that for every budget $T$,*

$$\mathbb{E}\big[\mathrm{GR}_T(\theta_i)\big] \ \geq \ \Delta/2.$$

*This impossibility holds whether the evaluator is stochastic or deterministic, since equation 11 is stated at the level of the full observation model $P_{e_{\mathrm{src}}}$.*

*Proof.* Let $\mathbb{P}_i$ be the distribution over the full transcript

$$\mathcal{T} := (p_{0:T-1}, y_{0:T-1}, \hat{p})$$

when the true hypothesis is $\theta_i$ and interaction is only with $e_{\mathrm{src}}$.

By equation 11, for any history and any chosen action $p_t$, the conditional distribution of $y_t$ is identical under $\theta_0$ and $\theta_1$. By induction on $t$, the entire transcript distribution is identical:

$$\mathbb{P}_0 = \mathbb{P}_1.$$

In particular, the marginal distribution of the final output $\hat{p}$ is the same under $\theta_0$ and $\theta_1$. Let this common distribution be denoted by $Q$ on $\mathcal{P}$.

Now consider the expected target regret under $\theta_0$: by equation 13, any output $\hat{p} \neq p_0$ incurs regret at least $\Delta$ under $\theta_0$:

$$\mathrm{GR}_T(\theta_0) = F_{e_{\mathrm{tgt}}}(p_0; \theta_0) - F_{e_{\mathrm{tgt}}}(\hat{p}; \theta_0) \ \geq \ \Delta \cdot \mathbf{1}\{\hat{p} \neq p_0\}.$$

Taking expectation w.r.t. $Q$ yields

$$\mathbb{E}[\mathrm{GR}_T(\theta_0)] \ \geq \ \Delta \cdot (1 - Q(\hat{p} = p_0)).$$

Similarly,

$$\mathbb{E}[\mathrm{GR}_T(\theta_1)] \ \geq \ \Delta \cdot (1 - Q(\hat{p} = p_1)).$$

Since $Q(\hat{p} = p_0) + Q(\hat{p} = p_1) \leq 1$, at least one of these probabilities is at most $1/2$, so at least one of the two expected regrets is at least $\Delta/2$:

$$\max\{\mathbb{E}[\mathrm{GR}_T(\theta_0)], \ \mathbb{E}[\mathrm{GR}_T(\theta_1)]\} \ \geq \ \Delta/2.$$

This proves the claim. $\qquad\square$

### C.3 CONCRETE EXAMPLES SATISFYING THE CONDITIONS

We give two illustrative examples where source data cannot distinguish two hypotheses, yet the target-optimal decision differs.

**Example 1: public test vs private (distribution shift / shortcut feature).** Let $\theta \in \{\theta_0, \theta_1\}$ encode which feature is truly stable/causal. Programs correspond to two model families: $p_0$ uses a stable causal feature; $p_1$ uses a shortcut feature. In the source environment (public benchmark), the shortcut feature is perfectly correlated with labels, so both hypotheses yield the same evaluator distribution for every program, satisfying equation 11. In the target environment (deployment/private), the shortcut correlation breaks: under $\theta_0$, $p_0$ is uniquely optimal; under $\theta_1$, $p_1$ is uniquely optimal, with margin $\Delta$, satisfying equation 13. No amount of interaction with $e_{\mathrm{src}}$ can identify which world holds.

Table 3: Mathematical definitions of auxiliary metrics across tasks. All metrics are deterministic outcome-level functionals of the program outputs. For subset-defined metrics (e.g., `large_circle_margin`), if the index set is empty, the metric value is defined as $0$.

| Task | Program Output | Aux Metric | Definition |
|---|---|---|---|
| Hadamard Matrix | $H \in \{\pm 1\}^{n \times n}$ binary matrix | row_orthogonality_deviation | $\frac{1}{n(n-1)} \sum_{i \neq j} \left\| \sum_k H_{ik} H_{jk} \right\|$ |
| | | row_sum_variance | $\mathrm{Var}\left( \sum_j H_{ij} \right)$ |
| | | element_balance | $\frac{1}{n^2} \sum_{i,j} \mathbf{1}[H_{ij} = +1]$ |
| | | log10_abs_det | $\log_{10} |\det(H)|$ |
| Second Autocorr Inequality | $f \in \mathbb{R}^n,\ f_i \geq 0$ nonnegative discrete function | smoothness_score | $\frac{1}{n-1} \sum_i |f_{i+1} - f_i|$ |
| | | center_concentration | $\sum_{|x_i| \leq 0.5} f_i \ /\ \sum_i f_i$ |
| | | sparsity | $\frac{1}{n} \sum_i \mathbf{1}[f_i < \varepsilon]$ |
| | | peak_to_average_ratio | $\max_i f_i \ /\ \mathbb{E}[f]$ |
| | | tail_mass | $\sum_{|x_i| > 0.5} f_i \ /\ \sum_i f_i$ |
| | | entropy | $-\sum_i p_i \log p_i, \quad p_i = f_i / \sum_j f_j$ |
| Circle Packing | $\{(C_i, r_i)\}_{i=1}^N$ circle centers and radii | density_score | $\sum_i \pi r_i^2 \ /\ S^2$ |
| | | center_spread_index | $\frac{1}{N} \sum_i \|C_i - (S/2, S/2)\|_2$ |
| | | radius_std_normalized | $\mathrm{Std}(r) \ /\ \mathbb{E}[r]$ |
| | | neighbor_distance_ratio | $\frac{1}{N} \sum_i \min_{j \neq i} \|C_i - C_j\|_2 \ /\ r_i$ |
| | | large_circle_margin | $\frac{1}{|I|} \sum_{i \in I} \left( \min(C_i^x, S - C_i^x, C_i^y, S - C_i^y) - r_i \right),\ I = \{i : r_i > \mathbb{E}[r]\}$ |
| | | pairwise_radii_product_sum | $\sum_{i<j} r_i r_j$ |
| | | centroid_distance_variance | $\mathrm{Var}(\|C_i - \mathbb{E}[C]\|_2)$ |
| ADAS-AIME | $(\text{accuracy}, \text{cost}, df)$ evaluation records | boxed_format_rate | $100 \cdot \frac{1}{N} \sum_i \mathbf{1}[\text{response}_i \text{ contains } \boxed{\cdot}]$ |
| | | three_digit_answer_rate | $100 \cdot \frac{1}{N} \sum_i \mathbf{1}[\text{len}(\text{answer}_i) = 3]$ |
| | | cost_efficiency | $\text{accuracy} \ /\ \sum_i \text{cost}_i$ |
| | | accuracy_variance | $\mathrm{Var}(\text{accuracy over sliding windows})$ |
| | | max_consecutive_errors | $\max_k \sum_{t=t_k}^{t_k + \ell} \mathbf{1}[\text{incorrect}_t]$ |

**Example 2: relaxed verification (slack) vs exact verification (constraint shift).** In combinatorial optimization, it is common to evaluate candidate programs using a *relaxed* verifier (e.g., allowing numerical slack), then validate with an *exact* verifier. For instance, in circle packing, one may verify non-overlap with a numerical slack such as $10^{-6}$, and later validate with an exact checker; converting a relaxed-feasible solution into an exact-feasible one may require tiny but nonzero modifications, and rankings can change when switching verifiers. This is explicitly discussed in the context of circle packing verification with slack vs exact validation. The source environment $e_{\mathrm{src}}$ can correspond to the relaxed evaluator, while the target environment $e_{\mathrm{tgt}}$ corresponds to the exact evaluator. Then two hypotheses $\theta_0, \theta_1$ can be constructed so that they are indistinguishable under the relaxed evaluator for all queried programs, yet the exact evaluator reverses which program is truly best (with gap $\Delta$), matching Theorem C.1.

## D  MORE DETAILS ON OUTCOME-LEVEL FACTORS

