# OpenReview forum: "CausalEvolve: Towards Open-Ended Discovery with Causal Scratchpad"
_ICLR.cc/2026/Workshop/FM4Science — ICLR 2026 Workshop FM4Science Poster_

### Official Review · Reviewer_HwVH · 2026-02-14
**Enhancing AI Scientist via Causal Scratchpad: A Rigorous Framework for Sample-Efficient and Generalizable Scientific Discovery**

**Rating:** 8
**Confidence:** 4

**Review:**

**1. Summary:**

This paper proposes CausalEvolve, a novel framework designed to enhance LLM-based evolutionary agents (AI Scientists) for open-ended scientific discovery. The authors identify two critical limitations in current agents like AlphaEvolve: poor sample efficiency and limited generalizability due to a lack of causal understanding. By modeling the discovery process as a Partially Observable Markov Decision Process (POMDP) and introducing a Causal Scratchpad, the framework enables agents to identify outcome-level and procedure-level causal factors. Theoretically, the paper proves that incorporating causal structures can reduce sample complexity from O(K) to O(d log K). Empirically, CausalEvolve outperforms SOTA baselines (e.g., ShinkaEvolve) across four challenging tasks: Hadamard Matrix, Second Autocorrelation Inequality, Circle Packing, and AIME mathematical problem-solving.

**2. Pros**

- Originality and Motivation: The integration of Causal Inference with Evolutionary Computing in the context of LLM agents is highly timely. Addressing "spurious correlations" in scientific discovery is a significant step beyond simple prompt-based iteration.
- Theoretical Rigor: The paper provides solid mathematical foundations. Theorem 3.2 and Theorem 3.3 effectively formalize why causal knowledge is essential for both efficiency and robust generalization across environment shifts.
-  Methodological Innovation: The "Causal Scratchpad" is well-structured. The distinction between Outcome-level factors (for efficiency) and Procedure-level factors (for structural insights) provides a comprehensive view of the program evolution process.
-   Abductive Reasoning: The "surprise detection" module allows the agent to perform abductive reasoning when experimental results deviate from causal expectations, mimicking a human scientist's ability to refine hypotheses.
-    Strong Empirical Results: The tasks chosen are non-trivial and span discrete optimization, geometry, and symbolic reasoning. Using Grok-4.1-fast-reasoning as a backbone ensures the results are representative of current frontier LLM capabilities.

**3. Cons**

- Dependency on LLM Prior Knowledge: The "Factor Construction" phase relies heavily on the LLM's ability to propose meaningful factors and corresponding code. If the LLM's prior is biased or insufficient for a highly niche scientific domain, the causal guidance might become a bottleneck.
- ATE Estimation Sensitivity: As noted in Section 4.2, the Average Treatment Effect (ATE) estimation may be biased due to small sample sizes and hidden confounders. While the authors claim "order-preserved quantities" are sufficient, more analysis on how estimation error impacts evolution stability would be beneficial.
-  Computational Overhead: The framework introduces additional LLM calls for factor identification, ATE estimation, and abductive reasoning. A brief discussion or table comparing the total token cost or wall-clock time against baselines would improve the "efficiency" argument.
-  Ablation Complexity: While the authors compare against COAT and ShinkaEvolve variants, a deeper ablation on the "Abductive Reasoning" module specifically (e.g., CausalEvolve vs. CausalEvolve w/o Surprise Detection) would clarify its individual contribution.

**4. Questions & Suggestions to Authors**

- [ ] Robustness to Incorrect Factors: How does the system handle a situation where the LLM identifies a "hallucinated" or incorrect causal factor? Is the reward mechanism in the Causal Planner robust enough to discard such factors
- [ ] Generalization across LLMs: While Grok-4.1 was used for fairness, would the causal framework show similar relative gains on smaller models, or is a high level of "reasoning" capability a prerequisite?

---

### Official Review · Reviewer_BCVc · 2026-02-25
**Review of CausalEvolve**

**Rating:** 5
**Confidence:** 3

**Review:**

CausalEvolve aims to address oscillations and inefficiencies in standard LLM-based evolutionary agents by introducing causal reasoning to improve directed behavior. The authors model this targeted evolution as a partially-observable Markov Decision Process, and augment the LLM with a causal "scratch pad" that allows the agent to reason over different optimization properties, modelled as a Multi-arm Bandit. The paper then validates CausalEvolve against a set of SOTA models, namely, ShinkaEvolve, on a set of optimization tasks.

The paper is clearly written and motivates the methodological and theoretical additions well. The proofs are sound and support the methodological claims of the paper, specifically why causal reasoning is the only path to full generalization.

However, the motivation for the paper is to address inefficiencies and oscillations in LLMs, but this motivation is not actually validated. The main results table reports accuracies on a set of four optimization tasks, however it's unclear that evolutionary agents fail or oscillate on these tasks. AIME is a binary task which generally does not result in oscillatory behavior, and the circle packing task again is perhaps not complex enough to demonstrate optimality where traditional models fail. Additionally, only mean performance is reported in Table 1, which again does not allow for the reader to see the optimized trajectories of the models reasoning. Variance plots of reasoning over time would be necessary to justify the oscillation motivation.

Additionally, the paper introduces significant complexity and makes claims about the need for causality, but this claim is not justified in the results. The model performs similarly to non-causal models on these tasks, so it is unclear that the causal component is actually necessary to improve performance on these tasks.

If the authors were to justify causality by diving into an aspect of these tasks that is not abundantly clear from the paper, and demonstrate efficiency/non-oscillation, I would certainly increase my score.

---

### Official Review · Reviewer_oNLY · 2026-02-25
**Compelling motivation and theoretical formalization, with clarification needed on implementation**

**Rating:** 5
**Confidence:** 3

**Review:**

The authors propose an evolutionary AI Scientist framework termed "CausalEvolve" to improve open-ended scientific discovery by injecting "causal" guidance into the search process. This work is motivated by the gap that current evolutionary methods rely too heavily on stochastic exploration or simple correlations, leading to oscillation and inefficiency. Overall, the motivation is compelling and aligned with automated scientific discovery. However, several aspects of the causal framing, factor definitions, and empirical validation require clarification.

Pros:
1. Strong motivation. Framing scientific discovery as a POMDP, where latent causal mechanisms govern observable outcomes, moves the discussion beyond simple prompt engineering.
2. Clear theoretical formalization of the problem in Section 3.

Cons:
1. Causal implementation is unclear. Section 3 presents a formal causal framing of scientific discovery using an SCM-based POMDP formulation. However, it is unclear how this formal causal framework is concretely instantiated in CausalEvolve. In Section 4, “causal knowledge” appears to be operationalized through outcome-level factors (LLM-generated metrics over outputs) and procedure-level factors (code-level descriptors used to guide search), which resembles guided heuristic search rather than explicit causal modeling. How is the SCM formulation in Section 3 implemented in practice? Does the system maintain or estimate any explicit causal structure? How do the interventions described theoretically correspond to the factors?
2. Lack of proof for addressing the gap. The stated gap is that existing agents "suffer from
decreasing evolution efficiency and exhibit oscillatory behavior when approaching known performance boundaries," yet the experiments do not provide the necessary metrics to prove that these are solved. It's unclear how to quantify oscillatory behavior, measure sample efficiency formally, or isolate improvements in convergence dynamics. Higher performance at fixed checkpoints suggests improved effectiveness, but does not directly demonstrate reduced oscillation or increased stability near performance boundaries. Additional analysis of search trajectories or convergence behavior would better support the stated claim.
3. Outcome-level factor determination and generalizability. The paper states that for a given task, LLMs are prompted with the basic task description to define a set of outcome-based factors and an executable code to map program output to the factor value. However, it is unclear how these factors are determined in a principled way across different tasks and what the impact of selecting different initial factors might be. Without clearer documentation or robustness analysis, it is difficult to assess how generalizable the approach is across diverse scientific discovery tasks. A side note, Appendix D appears to contain a formatting issue (Page 22).
4. Formatting inconsistencies. There are minor formatting and citation inconsistencies (e.g., capitalization of author names in line 33, in-text reference style in line 36) that need to be fixed.

---

### Official Review · Reviewer_k4B6 · 2026-02-25
**Review to Submission14**

**Rating:** 8
**Confidence:** 4

**Review:**

This paper introduces CausalEvolve, an evolutionary coding agent for open scientific discovery that enhances existing frameworks by incorporating causal buffers. The authors first formalise programme evolution as a POMDP model incorporating latent scientific knowledge parameters, demonstrating the gap in sampling efficiency and generalisation capability between ‘causal’ structured agents and black-box benchmarks. Building upon this foundation, CausalEvolve introduces: (i) outcome-level factors and a multi-armed bandit-style CausalPlanner to guide the direction of programme inspiration extraction; (ii) process-level factors, employing large language models for inductive reasoning on ‘unexpected patterns’. Experiments across four challenging discovery tasks demonstrate significant improvements over ShinkaEvolve and ablation studies.

### Strengths
1. This paper features a well-developed conceptual framework and substantial theoretical underpinnings.
2. It takes the causal scratchpad as a concrete mechanism rather than a slogan.
3. They did a nontrivial empirical evaluation across diverse, genuinely hard tasks.

### Weaknesses
1. Experimental setup and agent details are sparse for AIME and other tasks. For the AIME agent, the paper does not specify:
    - The base scaffolding agent’s structure beyond “same as ShinkaEvolve”;
    - How many calls to Grok-4.1-FR per step/problem;
    - Whether external tools (e.g., calculators) are used or not.
2. Missing or weak discussion of limitations.

---

### Meta-Review · Area_Chair_rrPB · 2026-02-27

**Recommendation:** Accept (Poster)
**Confidence:** 4

**Metareview:**

The average review score is above 6, which means reviewers recommended an acceptance.

---

### Decision · Program_Chairs · 2026-03-03

Accept (Poster)